# LEARNING LARGE-SCALE KERNEL NETWORKS

## ABSTRACT

This paper concerns large-scale training of *Kernel Networks*, a generalization of kernel machines that allows the model to have arbitrary centers. We propose a scalable training algorithm – EigenPro 3.0 – based on alternating projections with preconditioned SGD for the alternating steps. This is the first linear space algorithm for training kernel networks, which enables training models with large number of centers. In contrast to classical kernel machines, but similar to neural networks, our algorithm enables decoupling the learned model from the training set. This empowers kernel models to take advantage of modern methodologies in deep learning, such as data augmentation. We demonstrate the promise of EigenPro 3.0 on several experiments over large datasets. We also show data augmentation can improve performance of kernel models.

## 1  INTRODUCTION

*Kernel Machines* are predictive models described by the non-parametric estimation problem,

$$\min_{f \in \mathcal{H}} \ L(f) := \frac{1}{2} \sum_{i=1}^{n} L(y_i, f(\boldsymbol{x}_i)) + \lambda \left\| f \right\|_{\mathcal{H}}^2, \tag{1}$$

where $\mathcal{H}$ is a reproducing kernel Hilbert space (RKHS), and $(X, \boldsymbol{y}) = \{\boldsymbol{x}_i, y_i\}_{i=1}^{n}$ are training samples. By the representer theorem Wahba (1990), the solution to this problem has the form,

$$\widehat{f}(\boldsymbol{x}) = \sum_{i=1}^{n} \alpha_i K(\boldsymbol{x}, \boldsymbol{x}_i) \in \mathcal{H}, \tag{2}$$

where $K$ is the reproducing kernel corresponding to $\mathcal{H}$. The weights $\boldsymbol{\alpha} = (\alpha_i) \in \mathbb{R}^n$ are chosen to fit $(X, \boldsymbol{y})$. For example, kernel ridge regression takes the square loss $L(u, v) = (u - v)^2$, and the weights $\boldsymbol{\alpha} \in \mathbb{R}^n$ of the learned model are the unique solution to the $n \times n$ linear system of equations,

$$(K(X, X) + \lambda \boldsymbol{I}_n)\boldsymbol{\alpha} = \boldsymbol{y}, \tag{3}$$

where $[K(X, X)]_{ij} = K(\boldsymbol{x}_i, \boldsymbol{x}_j)$ is the matrix of pairwise kernel evaluations between samples.

However, observe that the kernel machine from equation (2), is *strongly coupled* to the training set, i.e., predictions from a learned model require access to the entire training dataset. There is no explicit control on the model size, it is always the same as the size of the dataset $n$. Such a coupling is inconvenient from an engineering perspective, and limits the scalability to large datasets for inference as well as for training. For instance, when fresh training samples are available, a larger system of equations needs to be solved, from scratch, to retrain the model.

In contrast, neural networks are decoupled from the training set. In particular, a pretrained neural network can be finetuned without any access to the original dataset. This decoupling affords the practitioner tremendous flexibility and is crucial for large-scale learning.

Deep learning methodologies take advantage of this scalability. For example, data augmentation is a widely used training technique to boost performance of neural networks, see Shorten & Khoshgoftaar (2019) for a review. Here, we augment the training set with artificial samples, which are obtained via perturbations or transformations to the true samples. For kernel machines, data augmentation means increasing the size of the dataset, and hence implicitly also the model size. Hence, data augmentation is prohibitively expensive for learning standard kernel machines.

*Kernel Networks* generalize kernel machines by allowing the flexibility to choose arbitrary centers. Perhaps most importantly, this leads the learned model to be *decoupled* from the training set.

**Definition 1** (Kernel Network). Given a kernel $K(\cdot, \cdot)$, a set of *centers* $Z := \{\mathbf{z}_i\}_{i=1}^p$, and weights $\boldsymbol{\alpha} = (\alpha_i) \in \mathbb{R}^p$, a kernel network is a function $\boldsymbol{x} \mapsto f(\boldsymbol{x}; Z, \boldsymbol{\alpha})$ given by

$$f(\boldsymbol{x}; Z, \boldsymbol{\alpha}) = \sum_{i=1}^p \alpha_i K(\boldsymbol{x}, \mathbf{z}_i). \tag{4}$$

We refer to $p$ as the model size, since there are $p$ degrees of freedom for the predictor.

Note that by definition, kernel networks do not require access to the training set to make predictions. This helps inference as well as training when $p \ll n$, and enables models to be trained on large-scale datasets. This also provides explicit capacity control by choosing $p$. Such a control is lacking in classical kernel machines since the model size is always $n$.

Kernel networks are classically studied in machine learning in the form of RBF networks, which correspond to radial kernels $K(\boldsymbol{x}, \mathbf{z}) = \phi(\|\boldsymbol{x} - \mathbf{z}\|)$. RBF networks were introduced by Broomhead & Lowe (1988) as a function approximation technique. Like neural networks, they are universal approximators for functions in $L^p(\mathbb{R}^d)$, see Park & Sandberg (1993); Poggio & Girosi (1990). Our definition extends to all positive definite kernels. This extension allows using kernels like the Convolutional Neural Tangent Kernel, which is neither radial nor rotationally invariant, among others.

## 1.1 PRIOR WORK

In the case when $Z = X$, which corresponds to standard kernel machines, there exist several solvers Wang et al. (2019); Gardner et al. (2018a); van der Wilk et al. (2020). For certain special kernels Si et al. (2014) enable speed-ups depending on the scale hyperparameter.

Classical procedures to learn kernel networks in their full generality, are to plug-in the functional form of equation (4) to solve problem (1). For example for the square loss, the solution satisfies,

$$(K(X, Z)^\top K(X, Z) + \lambda K(Z, Z))\boldsymbol{\alpha} = K(X, Z)^\top \boldsymbol{y}, \tag{5}$$

where $K(X, Z) \in \mathbb{R}^{n \times p}$ is the pairwise kernel evaluation between data $\boldsymbol{x}_i$ and centers $\mathbf{z}_j$. Notice when $\lambda$ is small, the solution converges to $K(X, Z)^\dagger \boldsymbol{y}$, which involves the pseudo-inverse. For other loss functions, iterative methods such as gradient descent can be used for minimizing the objective in terms of the weights $\boldsymbol{\alpha}$. Several regularized ERM approaches have been studied, see Que & Belkin (2016) and Scholkopf et al. (1997) for a review and comparisons. These methods suffer from poorly conditioned matrices, which significantly limits their rate of convergence. See Figure 3 in the Appendix for a deeper discussion on this approach and a comparison with problem-specific solvers.

**Nyström approximation:** Kernel networks with $Z \subset X$ have been studied extensively following Williams & Seeger (2000). This is perhaps the predominant strategy for applying kernel machines at scale, in the general case when random feature are hard to compute. Methods such as NYTRO Camoriano et al. (2016), and FALKON Rudi et al. (2017) are designed to work with such models. These methods require quadratic memory in terms of the model size. For example Meanti et al. (2020) only train models with 100,000 centers. Scaling these methods to higher model sizes is memory intensive. While these methods were not designed to train kernel networks in their generality, they perform surprisingly well for this task in high dimensions, since the distribution of the centers often closely resembles the distribution of the data. However one must exercise caution for training general kernel networks using these methods, i.e., when $Z \not\subset X$.

**Random features model:** Decoupled models for kernel machines have been considered earlier, perhaps most elegantly in the Random Features framework by Rahimi & Recht (2007). However, it is not straightforward to find the correct distribution that yields a desired target kernel, since sampling from the Fourier measure is not always tractable in high dimensions, especially for kernels that are not rotation invariant.

**Gaussian Process:** In the literature on GPs, sparse GPs Titsias (2009) is similar to kernel networks considered above. These models have so-called *inducing points* that reduce the model complexity. While several follow-ups such as Wilson & Nickisch (2015) and Gardner et al. (2018b) have been applied in practice, they require quadratic memory in terms of the number of inducing points, preventing scaling to large models. Indeed the inducing points interpretation is perhaps the most useful in choosing 'good' centers for kernel networks.

**EigenPro:** (short for Eigenspace Projections) is an iterative algorithm for kernel regression, i.e., when $Z = X$. It solves the linear system (3) by taking advantage of the problem structure. The algorithm applies a preconditioned Richarson iteration Richardson (1911), based on projecting gradients to certain eigenspaces of $K(X, X)$. EigenPro 2.0 Ma & Belkin (2019) improved upon EigenPro 1.0 Ma & Belkin (2017) by reducing the computational and memory costs for the preconditioner, by applying a stochastic approximation for estimating, and projecting onto the relevant eigenspaces.

EigenPro 2.0 cannot solve the general problem of learning a kernel machine, i.e., when $Z \neq X$. Our extension EigenPro 3.0, proposed in this paper fills this gap.

## 1.2 MAIN CONTRIBUTIONS

We develop a scalable iterative algorithm for learning kernel networks with a low memory footprint. Our training algorithm is derived using alternating but separate eigenspace projection steps. Importantly, our preconditioning preserves the decoupling between the model and the training set. EigenPro serves as the basis for our approach, and we use the same form for the preconditioner for more general problem of learning kernel networks. Our algorithm requires an additional projection step neccessary for this problem, which can be solved by a decoupled instance of EigenPro.

The focus of this paper is the design of the algorithm for training kernel networks in full generality with a linear space complexity in terms of the model size. We omit generalization and optimization properties of the algorithm, and will consider them in follow-up works. As such, these methods are expected to converge and behave well with analyses from linear systems and convex quadratic optimization being directly applicable. Furthermore, our method converges to a consistent estimator if we consider a student-teacher setup with known model centers.

Some noteworthy highlights of our training algorithm – EigenPro 3.0 – are:

1. **Model decoupled from training data:** Our algorithm fully respects the decoupling between the model and training data, and allows for any configuration of model centers. In particular, we do not require any label information on the centers. The flexibility of a decoupled model allows us to apply data augmentation for learning kernel networks. We demonstrate a gain in performance with this approach. We can also train overparameterized kernel networks with $p > n$.

2. **Linear space complexity:** Our algorithm can run with $O(p)$ memory, and $O(p^2)$ computations per iteration. Consequently, we can handle very large model sizes with a potential to scale much further. For example in our numerical experiments, we have trained models with 512,000 degrees of freedom. To our knowledge, this is the first general kernel network of this size trained with $\leq 100$ GB RAM.

**Organization:** In Section 3, we derive a vanilla version of our algorithm as a function space projected preconditioned gradient descent, with a decoupled model-agnostic preconditioner. In Section 4, we introduce several stochastic approximations that make our algorithm much faster and makes it scalable to large-scale datasets. Section 5 demonstrate the scalability of our algorithm to large datasets and large models over several datasets. Proofs are relegated to the Appendix.

## 2 PRELIMINARIES AND NOTATION

In what follows, functions are lowercase letters $a$, sets are uppercase letters $A$, vectors are lowercase bold letters $\boldsymbol{a}$, matrices are uppercase bold letters $\boldsymbol{A}$, operators are caligraphic letters $\mathcal{A}$, spaces and subspaces are bolfdace caligraphic letters $\boldsymbol{\mathcal{A}}$. Subscripts to sets, vectors, matrices indicate size.

If $K$ is a reproducing kernel for an RKHS $\boldsymbol{\mathcal{H}}$, then we have

$$\langle a, K(\cdot, \boldsymbol{x})\rangle_{\boldsymbol{\mathcal{H}}} = a(\boldsymbol{x}) \quad \forall a \in \boldsymbol{\mathcal{H}}, \qquad \langle K(\cdot, \boldsymbol{x}), K(\cdot, \mathbf{z})\rangle_{\boldsymbol{\mathcal{H}}} = K(\boldsymbol{x}, \mathbf{z}) = K(\mathbf{z}, \boldsymbol{x}). \quad (6)$$

**Evaluations and kernel matrices:** The vector of evaluations of a function $f$ over a set $X = \{\boldsymbol{x}_i\}_{i=1}^n$ is $f(X) := (f(\boldsymbol{x}_i)) \in \mathbb{R}^n$. We denote the kernel matrices $K(X, Z) \in \mathbb{R}^{n \times p}$, $K(X, X) \in \mathbb{R}^{n \times n}, K(Z, Z) \in \mathbb{R}^{p \times p}$, and $K(Z, X) = K(X, Z)^\top$. Similarly, $K(\cdot, X) \in \boldsymbol{\mathcal{H}}^{1 \times n}$,

and $K(\cdot, Z) \in \boldsymbol{\mathcal{H}}^{1 \times p}$, and for a set $A = \{\boldsymbol{a}_i\}_{i=1}^k$, and a vector $\boldsymbol{\alpha} = (\alpha_i) \in \mathbb{R}^k$, we use the notation

$$K(\cdot, A)\boldsymbol{\alpha} := \sum_{i=1}^k K(\cdot, \boldsymbol{a}_i)\alpha_i \in \boldsymbol{\mathcal{H}}, \qquad K(\mathbf{z}, A)\boldsymbol{\alpha} := \sum_{i=1}^k K(\mathbf{z}, \boldsymbol{a}_i)\alpha_i \in \mathbb{R}. \qquad (7)$$

Finally, for an operator $\mathcal{A}$, a function $a$, and a set $A = \{\boldsymbol{a}_i\}_{i=1}^k$, by

$$\mathcal{A}\{a\}(A) := (b(\boldsymbol{a}_i)) \in \mathbb{R}^k \qquad \text{where} \quad b = \mathcal{A}(a), \qquad (8)$$

**Definition 2.** [Top-$q$ eigensystem] Let $\lambda_1 > \lambda_2 > \ldots > \lambda_n$, be the eigenvalues of a hermitian matrix $\boldsymbol{A} \in \mathbb{R}^{n \times n}$, i.e., for unit-norm $\boldsymbol{e}_i$, we have $\boldsymbol{A}\boldsymbol{e}_i = \lambda_i \boldsymbol{e}_i$. Then we call the tuple $(\Lambda_q, \boldsymbol{E}_q, \lambda_{q+1})$ the top-$q$ eigensystem, where $\Lambda_q = \operatorname{diag}(\lambda_1, \lambda_2, \ldots, \lambda_q) \in \mathbb{R}^{q \times q}$, and $\boldsymbol{E}_q = [\boldsymbol{e}_1, \boldsymbol{e}_2, \ldots, \boldsymbol{E}_q] \in \mathbb{R}^{n \times q}$.

**Fréchet derivative:** Given a function $J : \boldsymbol{\mathcal{H}} \to \mathbb{R}$, the Frech'et derivative of $J$ with respect to $f$ is a linear functional, denoted $\nabla_f J$, such that

$$\lim_{\|h\|_{\boldsymbol{\mathcal{H}}} \to 0} \frac{|J(f+h) - J(f) - \nabla_f J(h)|}{\|h\|_{\boldsymbol{\mathcal{H}}}} = 0. \qquad (9)$$

Since $\nabla_f J$ is a linear functional, it lies in the dual space $\boldsymbol{\mathcal{H}}^*$. Since $\boldsymbol{\mathcal{H}}$ is a Hilbert space, it is self-dual, whereby $\boldsymbol{\mathcal{H}}^* = \boldsymbol{\mathcal{H}}$. If $L$ is the square loss for a given dataset $(X, \boldsymbol{y})$, i.e., $L(f) := \frac{1}{2}\sum_{i=1}^n (f(\boldsymbol{x}_i) - y_i)^2$ we can apply the chain rule, and using equation (6), and the fact that $\nabla_f \langle f, g \rangle_{\boldsymbol{\mathcal{H}}} = g$, we get, that the Fréchet derivative at $f = f_0$ is,

$$\nabla_f L(f_0) = \sum_{i=1}^n (f_0(\boldsymbol{x}_i) - y_i)\nabla_f f(\boldsymbol{x}_i) = \sum_{i=1}^n (f_0(\boldsymbol{x}_i) - y_i)K(\cdot, \boldsymbol{x}_i) = K(\cdot, X)(f_0(X) - \boldsymbol{y}). \quad (10)$$

**Hessian operator:** The Hessian operator $\nabla_f^2 L : \boldsymbol{\mathcal{H}} \to \boldsymbol{\mathcal{H}}$ for the square loss is given by,

$$\mathcal{K} := \sum_{i=1}^n K(\cdot, \boldsymbol{x}_i) \otimes K(\cdot, \boldsymbol{x}_i), \quad \mathcal{K}\{f\}(\mathbf{z}) := \sum_{i=1}^n K(\mathbf{z}, \boldsymbol{x}_i)f(\boldsymbol{x}_i) = K(\mathbf{z}, X)f(X). \qquad (11)$$

Note that $\mathcal{K}$ is surjective on $\boldsymbol{\mathcal{X}}$, and hence invertible when restricted to $\boldsymbol{\mathcal{X}}$. Note that when $\boldsymbol{x}_i \stackrel{\text{i.i.d.}}{\sim} \mathbb{P}$, for some measure $\mathbb{P}$, the above summation, on rescaling by $\frac{1}{n}$, converges due to strong law as,

$$\lim_{n \to \infty} \frac{K\{f\}}{n} = \mathcal{T}_K\{f\} := \int K(\cdot, \boldsymbol{x})f(\boldsymbol{x})\,\mathrm{d}\mathbb{P}(\boldsymbol{x}), \qquad (12)$$

which is an integral operator. The following lemma relates the spectra of $\mathcal{K}$ and $K(X, X)$.

**Proposition 1** (Nyström extension). *For $1 \le i \le n$, let $\lambda_i$ be an eigenvalue of $\mathcal{K}$, and $\psi_i$ its unit $\boldsymbol{\mathcal{H}}$-norm eigenfunction, i.e., $\mathcal{K}\{\psi_i\} = \lambda_i \psi_i$. Then $\lambda_i$ is also an eigenvalue of $K(X, X)$. Moreover if $\boldsymbol{e}_i$, is its unit-norm eigenvector, i.e., $K(X, X)\boldsymbol{e}_i = \lambda_i \boldsymbol{e}_i$, we have,*

$$\psi_i = K(\cdot, X)\frac{\boldsymbol{e}_i}{\sqrt{\lambda_i}}. \qquad (13)$$

We review EigenPro 2.0 which is a closely related algorithm for kernel regression, i.e., when $Z = X$.

**Background on EigenPro** (short for Eigenspace Projections): Proposed in Ma & Belkin (2017), EigenPro 1.0 is an iterative solver for solving the linear system in equation (3) based on a preconditioned stochastic gradient descent in the Hilbert space,

$$f^{t+1} = f^t - \eta \cdot \mathcal{P}\{\nabla_f L(f^t)\}. \qquad (14)$$

Here $\mathcal{P}$ is a preconditioner. Due to its iterative nature, EigenPro can handle $\lambda = 0$ in equation equation (3), corresponding to the problem of kernel interpolation, since in that case, the learned model satisfies $f(\boldsymbol{x}_i) = y_i$ for all samples in the training set.

It can be shown that the following iteration in $\mathbb{R}^n$

$$\boldsymbol{\alpha}^{t+1} = \boldsymbol{\alpha}^{t+1} - \eta(\boldsymbol{I}_n - \boldsymbol{Q})(K(X, X)\boldsymbol{\alpha}^t - \boldsymbol{y}), \qquad (15)$$

emulates equation (14) in $\mathcal{H}$, see Lemma 3 in the Appendix. The above iteration is a preconditioned version of the Richardson iteration, Richardson (1911), with well-known convergence properties. Here, $\boldsymbol{Q}$ as a rank-$q$ symmetric matrix obtained from the top-$q$ eigensystem of $K(X, X)$, with $q \ll n$.

The preconditioner, $\mathcal{P}$ acts to flatten the spectrum of the Hessian $\mathcal{K}$. In $\mathbb{R}^n$, the matrix $\boldsymbol{I}_n - \boldsymbol{Q}$ has the same effect on $K(X, X)$. The largest stable learning rate is then $\frac{2}{\lambda_{q+1}}$ instead of $\frac{2}{\lambda_1}$. Hence a larger $q$, allows faster training when $\mathcal{P}$ is chosen appropriately.

EigenPro 2.0 proposed in Ma & Belkin (2019), applies a stochastic approximation for $\mathcal{P}$ based on the Nyström extension. We apply EigenPro 2.0 to perform an inexact projection step in our algorithm.

## 3  EigenPro 3.0: Projected preconditioned gradient descent

In this section we derive EigenPro 3.0exact-projection, a precursor to EigenPro 3.0, for learning a kernel network. This algorithm is based on a function space projected gradient method. However it does not scale well. In Section 4 we make it scalable by applying stochastic approximations, which finally yields EigenPro 3.0 (Algorithm 1).

We want to solve the following constrained infinite dimensional problem,

$$\underset{f}{\text{minimize}} \ L(f) = \sum_{i=1}^{n}(f(\boldsymbol{x}_i) - y_i)^2, \qquad \text{subject to} \quad f \in \boldsymbol{\mathcal{Z}} := \text{span}\left(\{K(\cdot, \mathbf{z}_j)\}_{j=1}^{p}\right). \quad (16)$$

Thus the learned model $f$ is a linear combination of functions $\{K(\mathbf{z}_j, \cdot)\}_{j=1}^{p}$, just like Definition 1. We will apply the function-space projected gradient method to solve this problem,

$$f^{t+1} = \text{proj}_{\boldsymbol{\mathcal{Z}}}\left(f^t - \eta\mathcal{P}\left\{\nabla_f L(f^t)\right\}\right), \qquad \text{where} \quad \text{proj}_{\boldsymbol{\mathcal{Z}}}(u) := \underset{f \in \boldsymbol{\mathcal{Z}}}{\arg\min} \ \|u - f\|_{\boldsymbol{\mathcal{H}}}^2, \quad (17)$$

where $\nabla_f L(f^t)$ is the Fréchet derivative at $f^t$ as given in equation (10), $\mathcal{P}$ is a preconditioning operator given in equation (25), $\eta$ is a learning rate, and $\text{proj}_{\boldsymbol{\mathcal{Z}}} : \boldsymbol{\mathcal{H}} \to \boldsymbol{\mathcal{Z}}$ is the projection operator that projects functions from $\boldsymbol{\mathcal{H}}$ onto the subspace $\boldsymbol{\mathcal{Z}}$.

**Remark 1.** Note that even though equation (17) is an iteration over functions which are infinite dimensional objects $\{f^t\}_{t \geq 0}$, we can represent this iteration in finite dimensions as $\{\boldsymbol{\alpha}^t\}_{t \geq 0}$, where $\boldsymbol{\alpha}_t \in \mathbb{R}^p$. To see this, observe that $f^t \in \boldsymbol{\mathcal{Z}}$, whereby we express it as,

$$f^t = K(\cdot, Z)\boldsymbol{\alpha}^t \in \boldsymbol{\mathcal{H}}, \qquad \text{for an } \boldsymbol{\alpha}^t \in \mathbb{R}^p. \quad (18)$$

Furthermore, the evaluation of the function $f^t$ above at $X$, is

$$f^t(X) = K(X, Z)\boldsymbol{\alpha}^t \in \mathbb{R}^n. \quad (19)$$

**Gradient:**   Due to equations (10) and (19) together, the gradient is given by the function,

$$\nabla_f L(f^t) = K(\cdot, X)(f^t(X) - \boldsymbol{y}) = K(\cdot, X)(K(X, Z)\boldsymbol{\alpha}^t - \boldsymbol{y}) \in \boldsymbol{\mathcal{X}} := \text{span}(\{K(\cdot, \boldsymbol{x}_i)\}_{i=1}^{n}). \quad (20)$$

Observe that the gradient does not lie in $\boldsymbol{\mathcal{Z}}$ and hence a step of gradient descent would leave $\boldsymbol{\mathcal{Z}}$, and the constraint is violated. This necessitates a projection onto $\boldsymbol{\mathcal{Z}}$. For finitely generated subspaces such as $\boldsymbol{\mathcal{Z}}$, the projection operation involves solving a finite dimensional linear system.

**$\boldsymbol{\mathcal{H}}$-norm projection:**   Functions in $\boldsymbol{\mathcal{Z}}$ can be expressed as $K(\cdot, Z)\boldsymbol{\theta}$. Hence we can rewrite the projection problem in equation (17) as a minimization in $\mathbb{R}^p$, with $\boldsymbol{\theta}$ as the unknowns. Observe that,

$$\underset{f}{\arg\min} \ \|f - u\|_{\boldsymbol{\mathcal{H}}} = \underset{f}{\arg\min} \ \|f - u\|_{\boldsymbol{\mathcal{H}}}^2 = \underset{f}{\arg\min} \ \langle f, f \rangle_{\boldsymbol{\mathcal{H}}} - 2\langle f, u \rangle_{\boldsymbol{\mathcal{H}}}$$

since $\|u\|_{\boldsymbol{\mathcal{H}}}^2$ does not affect the solution. Further, using $f = K(\cdot, Z)\boldsymbol{\theta}$, we can show that

$$\langle f, f \rangle_{\boldsymbol{\mathcal{H}}} - 2\langle f, u \rangle_{\boldsymbol{\mathcal{H}}} = \boldsymbol{\theta}^\top K(Z, Z)\boldsymbol{\theta} - 2\boldsymbol{\theta}^\top u(Z). \quad (21)$$

This yields a simple method to calculate the projection onto $\boldsymbol{\mathcal{Z}}$.

$$\text{proj}_{\boldsymbol{\mathcal{Z}}}\{u\} = \underset{f \in \boldsymbol{\mathcal{Z}}}{\arg\min} \ \|f - u\|_{\boldsymbol{\mathcal{H}}} = K(\cdot, Z)\widehat{\boldsymbol{\theta}} = K(\cdot, Z)K(Z, Z)^{-1}u(Z) \in \boldsymbol{\mathcal{Z}}, \quad (22)$$

$$\text{where} \quad \widehat{\boldsymbol{\theta}} = \underset{\boldsymbol{\theta} \in \mathbb{R}^p}{\arg\min} \ \boldsymbol{\theta}^\top K(Z, Z)\boldsymbol{\theta} - 2\boldsymbol{\theta}^\top u(Z) = K(Z, Z)^{-1}u(Z). \quad (23)$$

Notice that $\widehat{\boldsymbol{\theta}}$ above is linear in $u$, and $f^t(Z) = K(Z, Z)\boldsymbol{\alpha}^t$. Hence we have the following lemma.

**Algorithm 1** EigenPro 3.0

**Require:** Data $(X, \boldsymbol{y})$, centers $Z$, batch size $m$, Nyström size $s$,, preconditioner level $q$.
1: Fetch subsample $X_s \subseteq X$ of size $s$
2: $(\boldsymbol{E}, \Lambda) \leftarrow$ top-$q$ eigensystem of $K(X_s, X_s)$
3: $\boldsymbol{C} = K(Z, X_s)\boldsymbol{E}(\Lambda^{-1} - \lambda_{q+1}\Lambda^{-2})\boldsymbol{E}^\top \in \mathbb{R}^{p \times s}$
4: **while** Stopping criterion is not reached **do**
5:      Fetch minibatch $(X_m, \boldsymbol{y}_m)$
6:      $\boldsymbol{g}_m \leftarrow K(X_m, Z)\boldsymbol{\alpha} - \boldsymbol{y}_m$
7:      $\boldsymbol{h} \leftarrow K(Z, X_m)\boldsymbol{g}_m - \boldsymbol{C}K(X_s, X_m)\boldsymbol{g}_m$
8:      $\boldsymbol{\theta} \leftarrow$ EigenPro 2.0$(Z, \boldsymbol{h})$
9:      $\boldsymbol{\alpha} \leftarrow \boldsymbol{\alpha} - \frac{n}{m}\eta\,\boldsymbol{\theta}$
10: **end while**

**Algorithm 2** EigenPro 3.0 exact-projection

**Require:** Data $(X, y)$, centers $Z$, initialization $\boldsymbol{\alpha}^0$, preconditioning level $q$.
1: $(\boldsymbol{E}, \Lambda) \leftarrow$ top-$q$ eigensystem of $K(X, X)$
2: $\boldsymbol{Q} \leftarrow \boldsymbol{E}(\boldsymbol{I}_q - \lambda_{q+1}\Lambda^{-1})\boldsymbol{E}^\top \in \mathbb{R}^{n \times n}$
3: **while** Stopping criterion not reached **do**
4:      $\boldsymbol{g} \leftarrow K(X, Z)\boldsymbol{\alpha} - \boldsymbol{y}$
5:      $\boldsymbol{h} \leftarrow K(Z, X)(\boldsymbol{I}_n - \boldsymbol{Q})\boldsymbol{g}$
6:      $\boldsymbol{\theta} \leftarrow K(Z, Z)^{-1}\boldsymbol{h}$
7:      $\boldsymbol{\alpha} \leftarrow \boldsymbol{\alpha} - \eta\,\boldsymbol{\theta}$
8: **end while**

EigenPro 2.0$(Z, \boldsymbol{h})$ approximates $K(Z, Z)^{-1}\boldsymbol{h}$
See Table 1 for comparison of costs.

**Proposition 2** (Projection). *The projection step in equation* (17) *can be simplified as,*

$$f^{t+1} = f^t - \eta\, K(\cdot, Z)K(Z, Z)^{-1}\left(\mathcal{P}\left\{\nabla_f L(f^t)\right\}(Z)\right) \in \boldsymbol{\mathcal{Z}}. \tag{24}$$

Hence, in order to perform the update, we only need to know $\mathcal{P}\left\{\nabla_f L(f^t)\right\}(Z)$, which can be evaluated efficiently for a suitably chosen preconditioner.

**Data preconditioner agnostic to model:** Just like with usual gradient descent, the largest stable learning rate is governed by the largest eigenvalue of the Hessian of the objective in equation (16), which is given by equation (11). The preconditioner $\mathcal{P}$ in equation (17) acts to reduce the effect of a few large eigenvalues. We choose $\mathcal{P}$ given in equation (25), just like Ma & Belkin (2017).

$$\mathcal{P} := \mathcal{I} - \sum_{i=1}^{q}\left(1 - \frac{\lambda_{q+1}}{\lambda_q}\right)\psi_i \otimes \psi_i \quad : \boldsymbol{\mathcal{H}} \to \boldsymbol{\mathcal{H}}. \tag{25}$$

Recall from Section 2 that $\psi_i$ are eigenfunctions of the Hessian $\mathcal{K}$, characterized in Proposition 1. Note that this preconditioner is independent of $Z$. Since $\nabla_f L(f^t) \in \boldsymbol{\mathcal{X}}$, we only need to understand $\mathcal{P}$ on $\boldsymbol{\mathcal{X}}$. Let $(\Lambda_q, \boldsymbol{E}_q)$ be the top-$q$ eigensystem of $K(X, X)$, see Def. 2. Define the rank-$q$ matrix,

$$\boldsymbol{Q} = \boldsymbol{E}_q(\boldsymbol{I}_q - \lambda_{q+1}\Lambda_q^{-1})\boldsymbol{E}_q^\top \in \mathbb{R}^{n \times n}. \tag{26}$$

The following lemma outlines the computation involved in preconditioning.

**Proposition 3** (Preconditioner). *The action of $\mathcal{P}$ from equation* (25) *on functions in $\boldsymbol{\mathcal{X}}$ is given by,*

$$\mathcal{P}\left\{K(\cdot, X)\boldsymbol{a}\right\} = K(\cdot, X)(\boldsymbol{I}_n - \boldsymbol{Q})\boldsymbol{a}, \qquad \forall\, \boldsymbol{a} \in \mathbb{R}^m. \tag{27}$$

Since we know from equation (20) that $\nabla_f L(f^t) = K(\cdot, X)(K(X, Z)\boldsymbol{\alpha}^t - \boldsymbol{y})$, we have,

$$\mathcal{P}\left\{\nabla_f L(f^t)\right\}(Z) = K(Z, X)(\boldsymbol{I}_n - \boldsymbol{Q})(K(X, Z)\boldsymbol{\alpha}^t - \boldsymbol{y}). \tag{28}$$

The following lemma combines this with Proposition 2 to get the update equation from Algorithm 2.

**Lemma 1** (Algorithm 2 iteration). *The following iteration in $\mathbb{R}^p$ emulates equation* (17) *in $\boldsymbol{\mathcal{H}}$,*

$$\boldsymbol{\alpha}^{t+1} = \boldsymbol{\alpha}^t - \eta\, K(Z, Z)^{-1}K(Z, X)(\boldsymbol{I}_n - \boldsymbol{Q})(K(X, Z)\boldsymbol{\alpha}^t - \boldsymbol{y}). \tag{29}$$

Algorithm 2 does not scale well to large models and large datasets. We now propose stochastic approximations that drastically make it scalable to both large models as well as large datasets.

## 4   SCALING UP COMPUTATIONS VIA MULTIPLE STOCHASTIC APPROXIMATIONS

Algorithm 2 suffers from 3 main issues. It requires — (i) access to entire dataset of size $O(n)$ at each iteration, (ii) $O(n^2)$ memory to calculate the preconditioner $\boldsymbol{Q}$, and (iii) $O(p^3)$ for the matrix inversion corresponding to an exact projection. This prevents scalability to large $n$ and $p$.

|  |  | Computation | | Memory |
|---|---|---|---|---|
|  | setup | per iteration | | |
| EigenPro 3.0 | $sq^2$ | $mp + ms + qs + ps + T_{ep2}$ | | $s^2 + sm + M_{ep2}$ |
| EigenPro 3.0 exact-projection | $nq^2 + p^3$ | $np + nq$ | | $pn + n^2$ |
| FALKON | $p^3$ | $np$ | | $p^2$ |

Table 1: (**Order complexity analysis.**) Table 4 in the Appendix clearly states all symbols. Here $T_{ep2}$ is the time it takes to run EigenPro 2.0 for the approximate projection. In practice we only run 1 epoch of EigenPro 2.0 for large scale experiments for which $T_{ep2} = O(p^2)$. Similarly, $M_{ep2} = O(p)$ is the memory rquired for running EigenPro 2.0. We assume the cost of a single kernel evaluation is $O(1)$ and the number of targets is $O(1)$.

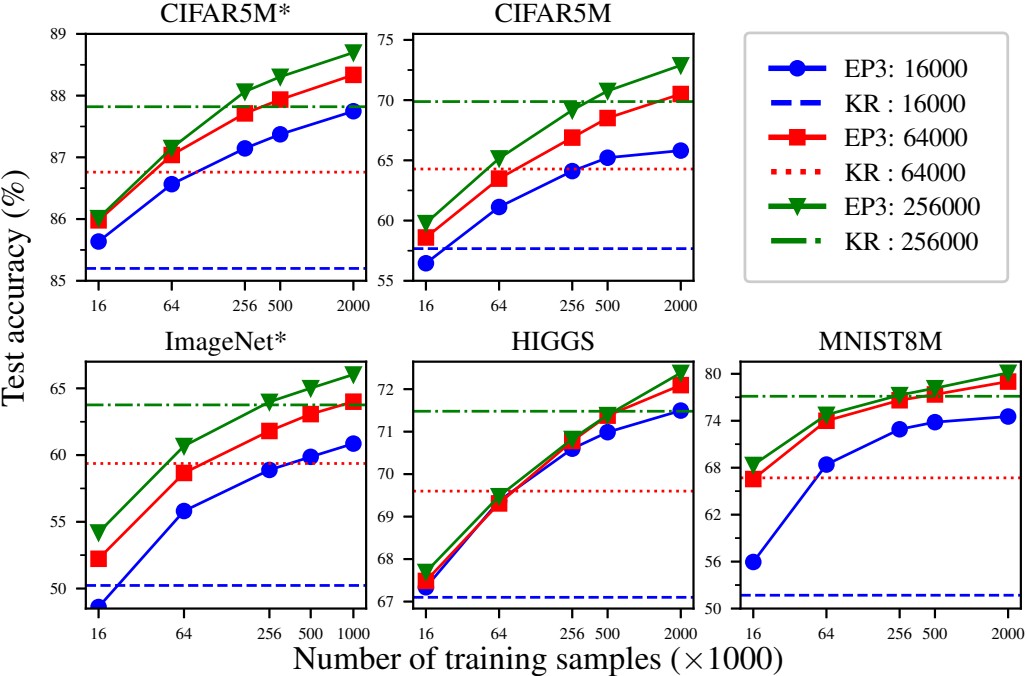

Figure 1: (**Large scale training.**) We randomly subsample to get model centers. The Kernel Regression (KR) baselines (dashed) are obtained from a standard kernel machine over the centers and their corresponding labels. Solid lines shows the performance of kernel networks trained using our algorithm, for varying number of training samples. Similar to neural networks, performance improves with more training samples. This is consistent across data sets.
* we applied a feature extraction before applying the kernel.

In this section we present 3 stochastic approximation schemes — stochsatic gradients, Nyström preconditioning, and inexact projection — that drastically reduce the computational cost and memory requirements. These approximations together give us Algorithm 1.

Algorithm 2 emulates equation (17), whereas Algorithm 1 is designed to emulate its approximation,

$$f^{t+1} = f^t - \frac{n}{m}\eta \cdot \widetilde{\text{proj}}_{\mathbb{Z}}\left(\mathcal{P}_s\left\{\widetilde{\nabla}_f L(f^t)\right\}\right), \tag{30}$$

where $\widetilde{\nabla}_f L(f^t)$ is a stochastic gradient obtained from a subsample of size $m$, $\mathcal{P}_s$ is a preconditioner obtained via a Nyström extension based preconditioner from a subset of size $s$, and $\widetilde{\text{proj}}_{\mathbb{Z}}$ is an inexact projection using EigenPro 2.0 to solve the projection equation $K(Z, Z)\boldsymbol{\theta} = \boldsymbol{h}$.

**Stochastic gradients:** We can replace the gradient with stochastic gradients, whereby $\widetilde{\nabla}_f L(f^t)$ only depends on a batch $(X_m, \boldsymbol{y}_m)$ of size $m$, denoted $X_m = \{\boldsymbol{x}_{i_j}\}_{j=1}^m$ and $\boldsymbol{y}_m = (y_{i_j}) \in \mathbb{R}^m$,

$$\widetilde{\nabla}_f L(f^t) = \sum_{j=1}^m (f(\boldsymbol{x}_{i_j}) - y_{i_j}) K(\cdot, \boldsymbol{x}_{i_j}) = K(\cdot, X_m)(K(X_m, Z)\boldsymbol{\alpha} - \boldsymbol{y}_m) \in \boldsymbol{\mathcal{X}}. \tag{31}$$

**Remark 2.** Here we need to scale the learning rate by $\frac{n}{m}$, to get unbiased estimates of $\nabla_f L(f^t)$.

**Nyström preconditioning:** We obtain an approximation for the preconditioner $\mathcal{P}$ from equation (25), which requires access to all samples. We use the Nystrom extension, see Williams & Seeger (2000). Consider a subset of size $s$, $X_s = \{\boldsymbol{x}_{i_k}\}_{k=1}^s \subseteq X$. We introduce the Nyström preconditioner,

$$\mathcal{P}_s := \mathcal{I} - \sum_{i=1}^s \left(1 - \frac{\lambda_{q+1}^s}{\lambda_i^s}\right) \psi_i^s \otimes \psi_i^s. \tag{32}$$

where $\psi_i^s$ are eigenfunctions of $\mathcal{K}^s := \sum_{k=1}^s K(\cdot, \boldsymbol{x}_{i_k}) \otimes K(\cdot, \boldsymbol{x}_{i_k})$. Note that $\mathcal{K}^s \approx \frac{s}{n}\mathcal{K}$ since both approximate $\mathcal{T}_K$ as shown in equation (12). However the scaling doesn't affect the preconditioner $\mathcal{P}_s$, since $\psi_i^s$ are unit norm. This preconditioner was first proposed in Ma & Belkin (2019).

Next, we need to understand the action of $\mathcal{P}_s$ on elements of $\boldsymbol{\mathcal{X}}$. Let $(\boldsymbol{E}_q, \Lambda_q)$ be the top-$q$ eigensystem of $K(X_s, X_s)$. Define the rank-$q$ matrix,

$$\boldsymbol{Q}_s := \boldsymbol{E}_q(\boldsymbol{I}_s - \lambda_{q+1}\Lambda_q^{-1})\Lambda_q^{-1}\boldsymbol{E}_q^\top \in \mathbb{R}^{s \times s}. \tag{33}$$

**Lemma 2** (Nyström preconditioning). *Let $\boldsymbol{a} \in \mathbb{R}^m$, and $X_m$ chosen like in equation (31), then,*

$$\mathcal{P}_s\{K(\cdot, X_m)\boldsymbol{a}\} = K(\cdot, X_m)\boldsymbol{a} - K(\cdot, X_s)\boldsymbol{Q}_s K(X_s, X_m)\boldsymbol{a}. \tag{34}$$

Consequently, using equation (31), we get,

$$\mathcal{P}_s\left\{\widetilde{\nabla}_f L(f^t)\right\}(Z) = \Big(K(Z, X_m) - K(Z, X_s)\boldsymbol{Q}_s K(X_s, X_m)\Big)\big(K(X_m, Z)\boldsymbol{\alpha}^t - \boldsymbol{y}_m\big) \in \mathbb{R}^p. \tag{35}$$

**Inexact projection:** The projection step in Algorithm 2 requires the inverse of $K(Z, Z)$ which is computationally expensive. However this step is solving the $p \times p$ linear system

$$K(Z, Z)\boldsymbol{\theta} = \Big(K(Z, X_m) - K(Z, X_s)\boldsymbol{Q}_s K(X_s, X_m)\Big)\big(K(X_m, Z)\boldsymbol{\alpha}^t - \boldsymbol{y}_m\big). \tag{36}$$

Notice that this is the kernel interpolation problem EigenPro 2.0 can solve. This leads to the update,

$$\boldsymbol{\alpha}^{t+1} = \boldsymbol{\alpha}^t - \frac{n}{m}\eta\,\widehat{\boldsymbol{\theta}}^T \qquad\qquad \text{(EigenPro 3.0 update)}$$

where $\widehat{\boldsymbol{\theta}}^T$ is the solution to equation (36) after $T$ steps of EigenPro 2.0 given in Algorithm 3 in the Appendix. Algorithm 1 implements the update above. Furthermore, EigenPro 2.0 itself applies a preconditioner which only depends on $Z$, no dependence on $X$, thus maintaining the decoupling.

**Remark 3** (Details on inexact-projection using EigenPro 2.0). We apply $T$ steps of EigenPro 2.0 for the approximate projection. This algorithm itself applies a fast preconditioned SGD to solve the problem. The algorithm needs no hyperparameters adjustment. More details in the Appendix.

**Complexity analysis:** We compare the order complexity of the run-time and memory requirement of Algorithm 1 before and after stochastic approximations with FALKON solver in Table 1.

## 5  NUMERICAL EXPERIMENTS

We perform experiments on these datasets: (1) CIFAR10, Krizhevsky et al. (2009), (2) CIFAR5M, Nakkiran et al. (2020), (3) ImageNet, Deng et al. (2009), (4) MNIST, LeCun (1998), (5) MNIST8M, Loosli et al. (2007), (6) Fashion-MNIST, Xiao et al. (2017), and (7) HIGGS, Baldi et al. (2014). All experiments are performed with the Laplacian kernel with a fixed bandwidth=5. In some cases, we perform a feature extraction using MobileNetv2 pretrained on the ImageNet dataset. The pre-trained model was obtained from Wightman (2019). Details on our implementation are in Appendix C. We treat $K$-class classification problems as $k$ independent regression problems with targets from $\{0, 1\}$.

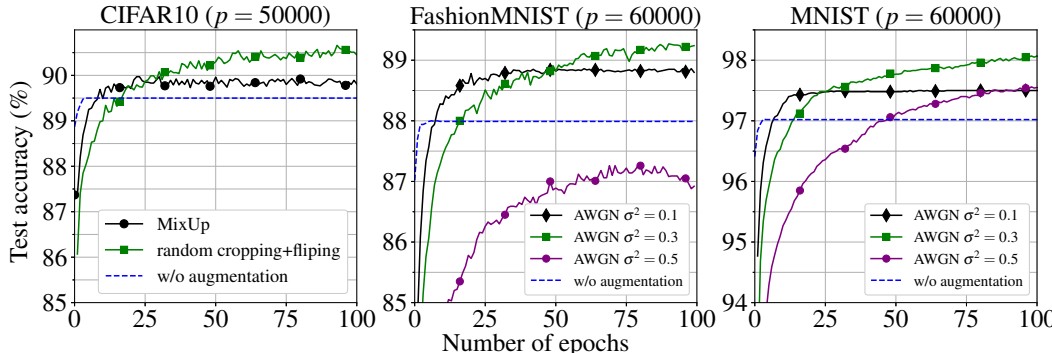

Figure 2: (**Data augmentation.**) We used the entire datasets as the model centers. We then trained the model on the augmented set. For CIFAR10 dataset, we apply MixUp and Crop+Flip augmentations, whereas for MNIST and FashionMNIST we add white gaussian noise with different variances. Performance improves by $\sim 1.5\%$ which is comparable to the gains seen by neural networks. The model without augmentation uses `EigenPro 2.0` to solve the kernel regression problem on the dataset.

**Large scale training:** In this experiment, we want to demonstrate we can learn models with 256,000 centers and 2,000,000 training samples. We also see that increasing the number of samples on a fixed model increases performance of the model. As a baseline, we compare with a standard kernel machine with the same model size $p$ (i.e., a kernel machine trained on the $p$ centers).

We apply our Algorithm 1 to train a model with $p$ centers which are a random subset of dataset with $n$ samples. We then systematically study the performance as we vary $n$ as well as $p$. Figure 1 shows that for a fixed model size adding more data improves performance significantly.

**Data augmentation:** Data augmentation is an important tool for enhancing the performance of deep networks. We demonstrate how this can improve performance of kernel models.

We conduct our experiment on CIFAR10 with feature extraction, raw images of MNIST and FashionMNIST. For CIFAR10 augmentation, we performed random cropping and flipping before feature extraction. Also for CIFAR10, we tried mix-up augmentation method in Zhang et al. (2017) after feature extraction. For MNIST and FashionMNIST augmentation we added Gaussian noise with different variances. In all cases, we used the entire training set as the centers and we generated augmented data set using the same training set. We performed 100 epochs for all of them. This means augmentation makes each data set effectively $\approx 100\times$ in size.

Figure 2 shows we have significant improvements in accuracy. We show results for the Laplacian kernel with a fixed bandwidth. We did not tune for the optimal bandwidth.

**Flexible choice of model centers:** In our model, the centers $z_i$ do not need to be a subset of the training samples. More importantly, the model is agnostic to labels at these points, in contrast to Falkon Rudi et al. (2017). In this experiment we show an example choice of centers that yields better performance than sub-sampling the dataset. We choose centers as the centroids from a K-means clustering procedure with $K = p$ using Omer (2020).

**Comparison to other works:** We demonstrate, in Table 3, that existing methods on center-based kernel regression for large data sets fail when number of centers are large. We compared our method with FALKON Rudi et al. (2017) and Gpytorch Gardner et al. (2018a). We used fixed 100GB RAM in all methods. For Gpytorch we tried to reproduce the version Rudi et al. (2017) used in their experiment, stochastic variational GPs (SVGP). We noticed they used SVGP with very small number of centers $\approx 2000$. We could not run Gpytorch with large number of centers. Also, note FALKON orginally did not have any notion of centers. They only used sub-sampling for more efficient computation. However, we found out that their code can also be used when centers are not a sub-set of the original data.

---

[1]Here we store the full $K(Z, Z)$ matrix on GPU. This is not possible for larger number of centers.

| Dataset | Model | Solver | $p = 100$ | $p = 1000$ | $p = 10000$ |
|---|---|---|---|---|---|
| CIFAR10 ($n = 50000$) | $k-$means | EP3.0 (ours) | **36.24** | **45.12** | **52.72** |
| | random | EP3.0 (ours) | $33.37 \pm 0.50$ | $44.19 \pm 0.09$ | $49.92 \pm 0.08$ |
| | random | FALKON | $34.16 \pm 0.36$ | $44.44 \pm 0.14$ | $50.40 \pm 0.12$ |
| CIFAR10 (MobileNetV2) ($n = 50000$) | $k-$means | EP3.0 (ours) | **82.69** | **86.58** | **89.11** |
| | random | EP3.0 (ours) | $74.29 \pm 0.44$ | $84.38 \pm 0.15$ | $86.58 \pm 0.06$ |
| | random | FALKON | $74.57 \pm 0.71$ | $84.81 \pm 0.14$ | $86.82 \pm 0.09$ |
| MNIST ($n = 60000$) | $k-$means | EP3.0 (ours) | **91.89** | **95.96** | **97.69** |
| | random | EP3.0 (ours) | $87.24 \pm 0.015$ | $94.96 \pm 0.102$ | $97.31 \pm 0.004$ |
| | random | FALKON | $88.43 \pm 0.604$ | $95.39 \pm 0.047$ | $97.64 \pm 0.080$ |
| FashionMNIST ($n = 60000$) | $k-$means | EP3.0 (ours) | **78.66** | **85.55** | 88.13 |
| | random | EP3.0 (ours) | $76.24 \pm 0.003$ | $84.59 \pm 0.069$ | $87.84 \pm 0.036$ |
| | random | FALKON | $77.33 \pm 0.631$ | $85.15 \pm 0.214$ | $\mathbf{88.27 \pm 0.093}$ |

Table 2: (**Flexible model.**) The separation between model and data empowers us to choose model centers which better represent the underlying data distribution. We compare two model types, and two solvers. The $k-$means model, as suggested by Que & Belkin (2016), chooses centers to be $k-$means of the training set, whereas the random model chooses a random subset to be the centers. Choosing centers as $k-$means improves the performance, especially when model size $p$ is considerably smaller than dataset size $n$.

| Model Size | EigenPro 3.0 | EigenPro 3.0[1] | FALKON | Gpytorch |
|---|---|---|---|---|
| $p = 32K$ (CIFAR5M) | 87.92% (727.78s ) | 87.93% (322.45s) | **88.04%** (**163.59s**) | — |
| $p = 64K$ (CIFAR5M) | **88.15**% (2178.8s ) | 88.14 % (515.94s) | 87.23% (**368.892s**) | — |
| $p = 128K$ (CIFAR5M) | **88.3**5% (**7453.772s**) | — | — | — |
| $p = 256K$ (CIFAR5M) | **88.46**% (**24386.28s** ) | — | — | — |
| $p = 512K$ (CIFAR5M) | **88.55**% (**71784.23s**) | — | — | — |

Table 3: (**Large model size.**) We preformed this experiment on extracted feature of CIFAR5M dataset. This shows that both Gpytorch and FALKON fail for large number of centers. Also, note that we are not using any tricks to optimize our algorithm for best time performance. However, if we could store the whole $K(Z, Z)$ matrix, third column, we can get a significant speed up.

## 6 CONCLUSION

Kernel networks, unlike kernel machines are a class of kernel models decoupled from the training dataset. In this paper we presented a fast and scalable training algorithm — EigenPro 3.0 — for learning general kernel networks on large scale datasets, in a manner that preserves the decoupling property of the learned model, and does not require label information at the model centers.

The method relies on alternating projection operations with preconditioning, one dependent only on the training data, while the other only on the model centers. We proposed stochastic approximations that make our algorithm scalable to large datasets as well as large model sizes. The algorithm has a linear space complexity in terms of the number of model centers and does not require any matrix inversion.

Through numerical experiments, we provided evidence that demonstrates the promise of this algorithm on several datasets from across problem domains. In particular, we showed kernel models can benefit from data augmentation without increasing the model complexity. Our algorithm enables various modern machine learning techniques for training kernel methods. The next step is to scale up the algorithm to train large models with millions of centers and billions of samples.

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
