# OpenReview forum: "Learning large-scale Kernel Networks"
_ICLR.cc/2023/Conference — Submitted to ICLR 2023_

### Official Review · Reviewer_nu3W · 2022-10-23

**Confidence:** 2
**Correctness:** 4
**Technical Novelty And Significance:** 2
**Empirical Novelty And Significance:** 3
**Recommendation:** 5

**Clarity, Quality, Novelty And Reproducibility:**

The paper is well-written; novelty is kind of marginal to me; reproducibility is unclear.

**Strength And Weaknesses:**

Strengths:
- The paper is well-written and well-motivated;
- The analysis is quite thorough and easy to follow.

Weakness:
- My major concern is that related works are not well-addressed. I think it is better to inlcude an "related work" section. I am not an experts in kernel machine, but I think there should be a lot of related work on kernal networks (Nyström approximate kernel networks). For example, see [1] and the references therein.

Some questions:
- There is not comparison on convergence speed betwen EigenPro 3.0 and other baseline methods? Is there a particular reason not to include this experiment?
- I am not very clear about the technical novelty of this paper compared to EigenPro 2.0 and EigenPro 1.0. It seems that EigenPro 3.0 uses similar technique as previous works (preconditioned gradient descent), the only difference is that the problem is kernel networks instead of the kernel machine. Is there any particular technical challenges of applying preconditioned gradient descent to kernel networks? Can the author(s) explain the technical contribution compared with EigenPro 2.0 and 1.0?

[1] Memory Efficient Kernel Approximation, Si et al. JMLR 2017.


**Summary Of The Paper:**

The author(s) proposed an preconditioned gradient descent algorithm for kernel networks. The algorithm is built on EigenPro 2.0. Experiments on real-world datasets are conducted to evaluate the proposed EigenPro 3.0.

**Summary Of The Review:**

Overall the paper is well-written and seems to be a descent contribution to the training of kernel networks. Related work is not well-addressed. I will raise my score if the author(s) can address the questions I raised and add more about discussion about related works.

---

> ### Author Response · Authors · 2022-11-18
> **Thank you for your feedback**
>
> General official comment:
>
> We thank the reviewers for their comments and suggestions. This has led us to make several improvements to the manuscript as listed below:
>
> 1. We have expanded the Prior Work section to include random feature models and gaussian process models. These models also have limitations that cannot solve the problem we consider in its full generality. Our method is general and can work with any positive definite kernel function.
> 2. We have now explained our Main Contributions clearly. Our main contribution is a linear space algorithm, as compared to quadratic space requirements by competing methods. As a result we can train models > 512,000 centers with limited hardware available to our research group. This can be scaled further to much larger models with more engineering effort. This paper serves as a proof of concept for training such models at scale.
> 3. Table 3 shows experiments with competing methods. Other competing methods run out of memory since they require p^2 memory for training a model with p centers.
>
> *regarding Theory*: Since the focus of the paper is to provide an algorithm, presenting its mathematical derivation is important. As a result we have deferred theoretical analysis of the algorithm to future work. These analyses will follow directly from standard tools in learning theory and optimization, and will guide the improvements to the selection of model and optimization hyperparameters. However we feel the paper is complete without such analyses due to our technical contributions and large scale experiments.
>
> Our code has automatic hyperparameter selection which adapts to the size of the problem. This is based on our theoretical analysis which we have omitted in the present paper to focus on the derivation of the algorithm.
>
> If the reviewer feels they have no outstanding vital concerns regarding our paper, we respectfully ask that they increase their score accordingly.
>
> Reviewer 4 official comment:
>
> We thank the reviewer for the comments and suggestions. This has led to several improvements in the paper, especially concerning the Prior works section. We hope that our rebuttal provides some clarifications.
>
> *regarding Prior work*: We have added new prior work which compares with competing methods that can train decoupled kernel models. We have also added Table 3 in the new manuscript comparing the training time with competing methods on limited hardware available to our group.
>
> *regarding Comparison to EigenPro(EP) 2.0*: EP2.0 is only able to train classical kernel machines. EP2.0 cannot train kernel networks, ie, when model centers are arbitrarily different from training samples. EP3.0 fills this gap and enables training kernel networks with a linear space complexity. We have now added a clear indication of this to the updated manuscript.
>
> *regarding Technical novelty*: The problem we solve is in full generality, ie, when model centers are completely arbitrary. This problem can only be solved when using preconditioning (see Figure 3 in the appendix for comparison to GD). The design of the preconditioner is non-trivial and is an important mathematical contribution of this paper. While it looks similar in form to EP2.0, it is significantly different in its derivation. We also justify why this is the best preconditioner when we are constrained with memory.
>
> *regarding Empirical novelty*: Our method is the first linear space algorithm for training kernel networks, compared to quadratic space requirements for all competing methods. We have added results in Table 3 which show comparisons using limited hardware accessible to our research group.

---

### Official Review · Reviewer_qfN8 · 2022-10-24

**Confidence:** 4
**Clarity, Quality, Novelty And Reproducibility:** The novelty of this work seems limite…
**Correctness:** 3
**Technical Novelty And Significance:** 1
**Empirical Novelty And Significance:** 3
**Recommendation:** 3

**Strength And Weaknesses:**

Strength:
It seems impressive to make kernel network work on datasets such as Cifar and Imagenet.

I have several concerns of this paper:

1. Novelty:
The novelty of the contribution seems quite limited. The idea of using data center instead of the whole training data in kernel network is not new. The main contribution seems using a stochastic gradient for training, however such approach is very standard and has quite limited novelty. I also believe such inexact approximation should have nice convergence property compared with the exact version since the approximation is unbiased (if the subset of preconditioner and functional gradient is chosen independently)?

2. What’s the advantage of kernel network in practice?
All the experiments used in this paper is typical benchmark used to evaluate deep learning model. And the reported performance is much weaker than neural network. I believe it would be nice to work on dataset that kernel network has its unique advantage over neural network rather than the dataset the deep learning model is so much stronger than other model. This helps to justify the necessity of studying the kernel network.

Question:
How do you select the center Z?


**Summary Of The Paper:**

This paper studies how to scale up the training/inference of kernel network. It uses data center to reduce the memory and inference cost. It proposes to use inexact stochastic approximation to approximate functional gradient and the preconditioner. Such stochastic gradient allows us to use data augmentation during the training that improves the performance.

**Summary Of The Review:**

The work is clearly presented. However, I have some major concerns over its novelty.

---

> ### Author Response · Authors · 2022-11-18
> **Thank you for your feedback**
>
> General official comment:
>
> We thank the reviewers for their comments and suggestions. This has led us to make several improvements to the manuscript as listed below:
> 1.  We have expanded the Prior Work section to include random feature models and gaussian process models. These models also have limitations that cannot solve the problem we consider in its full generality. Our method is general and can work with any positive definite kernel function.
> 2. We have now explained our Main Contributions clearly. Our main contribution is a linear space algorithm, as compared to quadratic space requirements by competing methods. As a result we can train models > 512,000 centers with limited hardware available to our research group. This can be scaled further to much larger models with more engineering effort. This paper serves as a proof of concept for training such models at scale.
> 3. Table 3 shows experiments with competing methods. Other competing methods run out of memory since they require p^2 memory for training a model with p centers.
>
> *regarding Theory*: Since the focus of the paper is to provide an algorithm, presenting its mathematical derivation is important. As a result we have deferred theoretical analysis of the algorithm to future work. These analyses will follow directly from standard tools in learning theory and optimization, and will guide the improvements to the selection of model and optimization hyperparameters. However we feel the paper is complete without such analyses due to our technical contributions and large scale experiments.
>
> Our code has automatic hyperparameter selection which adapts to the size of the problem. This is based on our theoretical analysis which we have omitted in the present paper to focus on the derivation of the algorithm.
>
> If the reviewer feels they have no outstanding vital concerns regarding our paper, we respectfully ask that they increase their score accordingly.
>
> Reviewer 3 official comment:
>
> We thank you for the review. We hope our rebuttal provides some clarifications.
>
> *regarding Novelty*: We have modified the introduction of the paper to clearly highlight the novelty and Main Contributions of our work. Our main contribution is (1) linear space complexity algorithm, compared to quadratic space complexity of all present day algorithms. (2) With limited resources, we can train models with very large sizes, please see Table 3 in the updated manuscript. (3) Design of preconditioner. Figure 3 in the appendix provides a comparison with standard techniques such as GD to solve this problem without preconditioning. In essence preconditioning is essential to learning. Our design of the preconditioner is novel and has not been derived before.
>
> *regarding Advantage of kernel networks*: Neural nets are notoriously difficult to train. Kernel networks on the other hand are easy to train, and far more stable. Kernel networks are rare in practice because before this work, no algorithm could train them with a small memory footprint. Our algorithm enables this. We believe that these models can compete with neural networks.
>
> *regarding Correctness*: It would really help us if you could point us towards the errors. To the best of our knowledge, the paper is technically correct. We are happy to clarify this in further discussions.

---

### Official Review · Reviewer_fRby · 2022-10-25

**Confidence:** 4
**Clarity, Quality, Novelty And Reproducibility:** The quality, clarity and originality …
**Correctness:** 2
**Technical Novelty And Significance:** 2
**Empirical Novelty And Significance:** 2
**Recommendation:** 3

**Details Of Ethics Concerns:**

I have no ethics concerns on this work.

**Strength And Weaknesses:**

Strength:
1. This paper considers the generated subspace instead of the original space and projection gradient methods to solve kernel regression based on data centers.
2. Iterative SGD methods are independent on the number of training examples. (same to EigenPro 2.0)
3. Preconditioning can reduce the iterative complexity. (same to EigenPro 2.0)
4. Nystroem methods further reduce the computation complexity per iteration. (same to EigenPro 2.0)

Weakness:
1. The title is rather confused, which is more related to conventional kernel methods rather than kernel networks. In my understanding, kernel networks usually stack kernels or random features.
2. The contribution of this paper is obscure. Some claimed contributions belong to prior work. It should clearly point between this work the the prior work.
3. In my opinion, the main contribution of this work lies in the incorporation between EigenPro 2.0 and projection gradient methods. However, this work doesn't emphasize the necessity of projection to the subspace and how to define the subspace.
4. Data centers or the subspace are rather import to this work, but it fails to introduce the difference between them to Nystroem centers and the sampling methods. If the data centers are sampled with data-dependent strategy, the sampling complexity is usually high, for example leverage score sampling. If the data centers are sampled uniformly, what's the benefits from data centers?
5. This work lacks the convergence analysis. Specifically, the convergence rate of preconditioning projection gradient methods, which is important to the overall computational complexity of the proposed algorithms.
6. This paper lacks generalization guarantees. As we known, the generalization error bounds of kernel ridge regression (KRR) have been well established. It have proven the minimax optimal rates for KRR. Since this work solves KRR with preconditioning SGD, it should also provide generalization guarantees. For example, the authors should analyze the generalization impacts from the number of data centers and the iteration number.

**Summary Of The Paper:**

This paper is more related to kernel methods rather than neural networks. This paper presented a preconditioned SGD methods based on alternating projections (data centers) for solving kernel (ridge) regression. And the authors further devise Nystr\"om based algorithm to reduce the required examples.

**Summary Of The Review:**

The paper has some merits, but overall, I think the contribution of this paper is limited based on the following reasons: 1) the novelty of algorithms are limited. This paper modified EigenPro 2.0 with projection gradient descent. 2) The work requires more discussion on the importance of data centers (subspace) and how to choose them. 3)This paper lacks essential convergence analysis and generalization guarantees.

---

> ### Author Response · Authors · 2022-11-18
> **Thank you for your feedback**
>
> General official comment:
>
> We thank the reviewers for their comments and suggestions. This has led us to make several improvements to the manuscript as listed below:
> 1. We have expanded the Prior Work section to include random feature models and gaussian process models. These models also have limitations that cannot solve the problem we consider in its full generality. Our method is general and can work with any positive definite kernel function.
> 2. We have now explained our Main Contributions clearly. Our main contribution is a linear space algorithm, as compared to quadratic space requirements by competing methods. As a result we can train models > 512,000 centers with limited hardware available to our research group. This can be scaled further to much larger models with more engineering effort. This paper serves as a proof of concept for training such models at scale.
> 3. Table 3 shows experiments with competing methods. Other competing methods run out of memory since they require p^2 memory for training a model with p centers.
>
> *regarding Theory*: Since the focus of the paper is to provide an algorithm, presenting its mathematical derivation is important. As a result we have deferred theoretical analysis of the algorithm to future work. These analyses will follow directly from standard tools in learning theory and optimization, and will guide the improvements to the selection of model and optimization hyperparameters. However we feel the paper is complete without such analyses due to our technical contributions and large scale experiments.
>
> Our code has automatic hyperparameter selection which adapts to the size of the problem. This is based on our theoretical analysis which we have omitted in the present paper to focus on the derivation of the algorithm.
>
> If the reviewer feels they have no outstanding vital concerns regarding our paper, we respectfully ask that they increase their score accordingly.
>
> Reviewer 2 official comment:
>
> We thank the reviewer for their comments and evaluation. Your review has led to several improvements in the manuscript. We hope our rebuttal provides some clarifications below:
>
> *regarding Comparison to EigenPro(EP) 2.0*: EP2.0 is only able to train classical kernel machines. EP2.0 cannot train kernel networks, ie, when model centers are arbitrarily different from training samples. EP3.0 fills this gap and enables training kernel networks with a linear space complexity. We have now added a clear indication of this to the updated manuscript.
>
> *regarding Title*: You correctly pointed out that our work is about kernel methods rather than neural networks. The word “kernel” is, unfortunately, overloaded in ML literature. However, we strongly feel that ‘Kernel Networks’ is the most faithful representation for the models we consider because they generalize RBF networks, a well-studied class of models, to arbitrary kernels.
>
> *regarding Contributions* - In the updated manuscript, we have now made our contributions clear in the introduction so that it delineates from prior work. Our main contributions are:
> (1) linear space complexity algorithm, compared to quadratic space complexity of all present day algorithms. (2) With limited resources, we can train models with very large sizes, please see Table 3 in the updated manuscript. (3) Design of preconditioner. Figure 3 in the appendix provides a comparison with standard techniques such as GD to solve this problem without preconditioning. In essence preconditioning is essential to learning. Our design of the preconditioner is novel and has not been derived before.
>
> *regarding Model centers* - Our concern in this paper is about scalable training of a kernel network when the centers are known. Indeed, finding good centers is an important and challenging problem, but it is beyond the scope of this paper. We have added a note about this in the introduction in Main Contributions.
>
> *regarding Theoretical analysis* - We have purposely omitted the convergence and generalization analysis since it distracts from the main focus of the paper which is the design of a scalable iterative training algorithm which requires only linear space, as against quadratic space required by any contemporary method. Our model can be scaled to very large models which is not possible for any other method.
> The convergence analyses, both optimization and generalization are important but we will present them in follow-up work.
>
> *regarding Correctness*: Your remark on the correctness is unclear. To the best of our knowledge everything presented in the original manuscript is accurate. It would really help us if you could point us towards the errata. We are happy to discuss more on this point.
>
> *regarding Empirical novelty*: Our method is the first linear space algorithm for training kernel networks. We have added results in Table 3 which show comparisons with competing methods.

---

### Official Review · Reviewer_ner6 · 2022-10-28

**Confidence:** 3
**Correctness:** 4
**Technical Novelty And Significance:** 3
**Empirical Novelty And Significance:** 2
**Recommendation:** 5

**Clarity, Quality, Novelty And Reproducibility:**

- Clarity: As discussed above, the clarity could be improved.
- Quality: The quality of the work is high.
- Novelty: I believe the contribution is novel. In a narrow sense, it is the first work to show how to adapt EigenPro2.0 to the “kernel network” setting. In a broader sense, I’m curious if this is the first method to combine (1) a kernel model decoupled from the training set with (2) an efficient learning algorithm that leverages pre-conditioning. I do not believe this is the case (see, for example [1], [2]), though I would need to perform a more careful literature review.
- Reproducibility: I believe the work is reproducible. Code is provided in the supplementary materials.

[1] Avron et al. Random Fourier Features for Kernel Ridge Regression: Approximation Bounds and Statistical Guarantees. ICML, 2017.

[2] Frangella et al. Randomized Nystrom Precondition. ArXiv, 2021.


**Strength And Weaknesses:**

Strengths:
- The algorithmic contribution is non-trivial: the paper presents an efficient way of performing kernel network learning with pre-conditioning. The pre-conditioning is important as it can significantly reduce the number of SGD steps necessary to learn a model (as discussed in the EigenPro and EigenPro 2.0 papers).
- I agree with the paper that it is very important to decouple the size of a kernel model from the size of the dataset used to train it, thus allowing the model to scale to very large training sets (with efficient training/inference), and use methods like data augmentation. Adapting pre-conditioning methods to this setting is an important contribution.
- The empirical results are relatively interesting, and demonstrate the important of the above-mentioned “decoupling”.

Weaknesses:
- It seems to me that the paper neglects to discuss (and compare with) other efficient methods for kernel learning which decouple the model from the training set. For example, the Nystrom method (Williams & Seegar, 2001) can be used with arbitrary model centers from the input space; Zhang and Kwok (2010) showed that using k-means on the training set is an effective way of choosing the model centers for Nystrom. As another example, random Fourier features (Rahimi and Recht, 2007) also decouple the model from the training set, making it trivial to use methods like data augmentation together with large-scale kernel learning. While it is still possible that EigenPro3.0 is in many ways better (e.g., more efficient training, more compact models, better generalization, etc.) than these existing methods, these comparisons have not been explored at all, and thus the various trade-offs between these methods are not currently understood.
- The paper is often quite hard to follow. It could be beneficial to move most of the math related to functional spaces and Frechet derivatives to the appendix, and focus on the finite dimensional optimization problems instead. It could also be beneficial to describe EigenPro2.0 in more detail, especially because it is directly invoked from the EigenPro3.0 algorithm (line 8 of Algorithm 1). Simplifying the mathematical presentation as much as possible would be valuable (I’m aware this is non-trivial, given the complexity of the method, but I still think the presentation could be significantly improved).
- Currently, no theoretical results are provided about EigenPro3.0. Can the convergence/generalization properties of this algorithm be analyzed?


**Summary Of The Paper:**

This paper presents EigenPro3.0, an efficient algorithm for learning large scale kernel networks. EigenPro3.0 can be seen as an extension of the EigenPro2.0 method (Ma and Belkin, 2019) to the case where the function being learning is a “kernel network” instead of a “kernel machine”; a “kernel network” is a function of the form $f(x) = \sum_{i=1}^p a_i k(x, x_i)$, where the set of “centers” $\\{x_i\\}_{i=1}^p$ is allowed to be an arbitrary set of points in the input space X (as opposed to having to equal the training set, which is the case for “kernel machines”). Importantly, “kernel networks” decouple the size of the model from the size of the training set, which allows for efficiently applying methods like data augmentation to kernel learning.

Empirically, the paper demonstrates that:
1. When using a fixed number of centers $p$, performance improves as the amount of training data $n$ grows larger than $p$ (thus “kernel networks” are better than “kernel machines”), and
2. Applying data augmentation to kernel learning (while keeping the model centers equal to the unaugmented training set) leads to better performance than simply learning with the unaugmented dataset (again showing that “kernel networks” are better than “kernel machines”, as they allow applying data augmentation without increasing the model size).


**Summary Of The Review:**

This work presents a nice extension of EigenPro2.0 to the “kernel network” setting, where kernel model size is decoupled from the training set size. While this is a very nice technical contribution, an important limitation of the paper in its current form is that it does not compare to other large-scale kernel learning methods which also decouple the model size from the training set size (e.g., Nystrom, Random Fourier Features), thus making it hard to fully evaluate this work. Thus, I lean toward rejecting this paper, and recommending it be resubmitted to a future venue once discussion and comparison with related work is provided.

---

> ### Author Response · Authors · 2022-11-18
> **Thank you for your feedback**
>
> General official comment:
>
> We thank the reviewers for their comments and suggestions. This has led us to make several improvements to the manuscript as listed below:
> 1. We have expanded the Prior Work section to include random feature models and gaussian process models. These models also have limitations that cannot solve the problem we consider in its full generality. Our method is general and can work with any positive definite kernel function.
> 2. We have now explained our Main Contributions clearly. Our main contribution is a linear space algorithm, as compared to quadratic space requirements by competing methods. As a result we can train models > 512,000 centers with limited hardware available to our research group. This can be scaled further to much larger models with more engineering effort. This paper serves as a proof of concept for training such models at scale.
> 3. Table 3 shows experiments with competing methods. Other competing methods run out of memory since they require p^2 memory for training a model with p centers.
>
> *regarding Theory*: Since the focus of the paper is to provide an algorithm, presenting its mathematical derivation is important. As a result we have deferred theoretical analysis of the algorithm to future work. These analyses will follow directly from standard tools in learning theory and optimization, and will guide the improvements to the selection of model and optimization hyperparameters. However we feel the paper is complete without such analyses due to our technical contributions and large scale experiments.
>
> Our code has automatic hyperparameter selection which adapts to the size of the problem. This is based on our theoretical analysis which we have omitted in the present paper to focus on the derivation of the algorithm.
>
> If the reviewer feels they have no outstanding vital concerns regarding our paper, we respectfully ask that they increase their score accordingly.
>
> Reviewer 1 official comment:
>
> We thank the reviewer for their comments and evaluation. Your review has led to several improvements in the manuscript, especially the prior works section. We hope our rebuttal provides some clarifications below.
>
> *regarding Prior Work*: We have updated the prior works section of the paper with clear comparisons to these methods and their limitations. We thank the reviewer for pointing out that random feature models also possess the decoupling property. A deeper literature review has also led us to so-called Sparse GPs with inducing points which possess this decoupling property. We have discussed these in the updated manuscript. However these methods also have limitations which we have now addressed. We have also updated the Main Contributions subsection to clearly highlight the contributions of this paper.
>
> *regarding Theory*: The focus of this paper is on the design of the algorithm and the quantities involved that enable it to scale to very large models and very large datasets. We defer the theoretical analyses to future works since we expect these will be straightforward to derive from standard learning theoretic and optimization tools. However in the present form, we feel the paper is complete without these analyses.
>
> *regarding Exposition*: We have tried our best to condense the mathematical derivations and make them accessible to the broad audience of ICLR. However since this is an important technical contribution of this paper, we hope it justifies keeping this in the main body.
>
> *regarding Empirical novelty*: Our method is the first linear space algorithm for training kernel networks, compared to quadratic space requirements for all competing methods. We have added results in Table 3 which show comparisons using limited hardware accessible to our research group.

---

### Decision · Program_Chairs · 2023-01-20

**Decision:**

Reject

**Justification For Why Not Higher Score:**



**Justification For Why Not Lower Score:**



**Metareview: Summary, Strengths And Weaknesses:**

In this paper the authors present a preconditioned gradient descent technique (called EigenPro 3.0) for kernel networks. Unfortunately, none of the reviewers suggested the submission to be accepted (due to its unclear writing and fuzzy contributions, unclear novelty, lack of comparison with existing large-scale kernel methods, lack of theoretical guarantees).